# Missense variants in human ACE2 strongly affect binding to SARS-CoV-2 Spike providing a mechanism for ACE2 mediated genetic risk in Covid-19: A case study in affinity predictions of interface variants

**Stuart A. MacGowan**[1⊕], **Michael I. Barton**[2⊕], **Mikhail Kutuzov**[2], **Omer Dushek**[2], **P. Anton van der Merwe**[2]*, **Geoffrey J. Barton**[1]*

**1** Division of Computational Biology, School of Life Sciences, University of Dundee, Dow Street, Dundee, Scotland, United Kingdom, **2** Sir William Dunn School of Pathology, South Parks Road, University of Oxford, Oxford, Oxfordshire, United Kingdom

⊕ These authors contributed equally to this work.
* anton.vandermerwe@path.ox.ac.uk (PAvdM); g.j.barton@dundee.ac.uk (GJB)

## Abstract

SARS-CoV-2 Spike (Spike) binds to human angiotensin-converting enzyme 2 (ACE2) and the strength of this interaction could influence parameters relating to virulence. To explore whether population variants in ACE2 influence Spike binding and hence infection, we selected 10 ACE2 variants based on affinity predictions and prevalence in gnomAD and measured their affinities and kinetics for Spike receptor binding domain through surface plasmon resonance (SPR) at 37°C. We discovered variants that reduce and enhance binding, including three ACE2 variants that strongly inhibited (p.Glu37Lys, $\Delta\Delta G = -1.33 \pm 0.15$ kcal mol$^{-1}$ and p.Gly352Val, predicted $\Delta\Delta G = -1.17$ kcal mol$^{-1}$) or abolished (p.Asp355Asn) binding. We also identified two variants with distinct population distributions that enhanced affinity for Spike. ACE2 p.Ser19Pro ($\Delta\Delta G = 0.59 \pm 0.08$ kcal mol$^{-1}$) is predominant in the gnomAD African cohort (AF = 0.003) whilst p.Lys26Arg ($\Delta\Delta G = 0.26 \pm 0.09$ kcal mol$^{-1}$) is predominant in the Ashkenazi Jewish (AF = 0.01) and European non-Finnish (AF = 0.006) cohorts. We compared ACE2 variant affinities to published SARS-CoV-2 pseudotype infectivity data and confirmed that ACE2 variants with reduced affinity for Spike can protect cells from infection. The effect of variants with enhanced Spike affinity remains unclear, but we propose a mechanism whereby these alleles could cause greater viral spreading across tissues and cell types, as is consistent with emerging understanding regarding the interplay between receptor affinity and cell-surface abundance. Finally, we compared mCSM-PPI2 $\Delta\Delta G$ predictions against our SPR data to assess the utility of predictions in this system. We found that predictions of decreased binding were well-correlated with experiment and could be improved by calibration, but disappointingly, predictions of highly enhanced binding were unreliable. Recalibrated predictions for all possible ACE2 missense variants at the Spike interface were calculated and used to estimate the overall burden of ACE2 variants on Covid-19.

**Data Availability Statement:** Data are available in the manuscript. In addition, this study employed several public datasets, which are available from their original sources and all necessary identifiers and accessions are provided in the methods. Data that was compiled and calculated as part of this study will be available from the BioStudies database (https://www.ebi.ac.uk/biostudies) under accession S-BSST649. Software is available from GitHub: https://github.com/bartongroup/covid19-ace2-variants.

**Funding:** GJB and SAM received support from the Biotechnology and Biological Sciences Research Council (BBSRC; https://bbsrc.ukri.org; grant codes: BB/J019364/1 and BB/R014752/1). GJB also received support from a Wellcome Trust Biomedical Resources Grant (Grant code: 101651/Z/13/Z). OD is supported by a Wellcome Trust Senior Fellowship in Basic Biomedical Sciences (207537/Z/17/Z828). The funders had no role in study design, data collection and analysis, decision to publish, or preparation of the manuscript.

**Competing interests:** I have read the journal's policy and the authors of this manuscript have the following competing interests: AvdM declares ownership of shares in BioNTech SE.

## Author summary

One of the first things the SARS-CoV-2 virus does to invade human cells is bind to a cell surface receptor called angiotensin-converting enzyme 2 (ACE2). The virus attaches to this receptor through its Spike protein and knowledge from other viruses tells us that the strength of this interaction influences how infectious and or virulent it is. We hypothesised that the Spike-ACE2 affinity might vary in people who have different amino acids in the part of ACE2 where Spike binds and consequently might be protected–or more at risk–from the virus. To test this idea, we measured the affinity of several ACE2 mutants, representing different versions found in humans, for the Spike protein and we found that some strengthened the interactions alongside others that weakened it. Most of these variants are rare, but two are present in over 1 in 1,000 individuals in certain populations and so might be important for the epidemiology of COVID-19. We then used computational methods to predict the affinity of even more ACE2 mutants than we could test in the lab and again found many that might alter this interaction. These data may help identify people who are at higher or lower risk from COVID-19.

## 1. Introduction

The COVID-19 pandemic is one of the greatest global health challenges of modern times. Although the disease caused by the severe acute respiratory syndrome coronavirus 2 (SARS-CoV-2) is usually cleared following mild symptoms, it can progress to serious illness and death [1]. Besides the clear risks associated with age and comorbidities [2,3], there could be a genetic component that predisposes some individuals to worse outcomes [4]. Genetic association studies have already identified several loci involved in Covid-19 risk [5]. Identifying further genetic factors of COVID-19 susceptibility has implications for clinical decision making and epidemic dynamics. Genetic variation may constitute hidden risk factors and, in some cases, explain why otherwise healthy individuals in low-risk groups experience severe disease. The identification of specific genetic variants that influence the severity and progression of COVID-19 presents the opportunity for predictive diagnostics, early intervention and personalised treatments whilst the population distribution of such variants could contribute to population specific risk.

Human angiotensin-converting enzyme 2 (ACE2) is the host cell receptor that SARS-CoV-2 exploits to infect human cells [1,6]. As this is the same receptor used by the SARS coronavirus (SARS-CoV) that caused the SARS outbreak in 2002, the detailed body of knowledge built around SARS-CoV infection is relevant to understanding SARS-CoV-2 [1,6,7]. The spike glycoprotein (Spike) is the coronavirus entity that recognises and binds host ACE2. Both SARS coronavirus Spikes include an S1 domain that contains ACE2 recognition elements and an S2 domain that is responsible for membrane fusion [6]. Spike is primed for cell fusion by cleavage with host furin [8] and TMPRSS2 [6], in SARS-CoV cleavage by TMPRSS2 is thought to be promoted upon formation of the ACE2 Spike complex [9]. The S1 receptor binding domains (RBDs) from both SARS-CoV [10] and SARS-CoV-2 [11] have been co-crystallised with human ACE2. The RBDs from both viruses are similar in overall architecture and interface with roughly the same surface on ACE2. Differences are apparent in the so-called receptor binding motif, which is the region of the RBD responsible for host range and specificity of coronaviruses [10–12]. The binding affinity of Spike and ACE2 is known to be correlated to the infectivity of SARS-CoV and is determined by the complementarity of the interfaces

[10,12]. However, despite its essential role in infection, risk variants in ACE2 have not been conclusively identified in genetic association studies.

Missense variants located in protein-protein interaction interfaces can affect altered binding characteristics [13,14] and in the context of virus-host interactions, have been shown to effect susceptibility [15]. This indicates the potential of missense variants in ACE2 to alter Spike binding and therefore influence a key step in SARS-CoV-2 infection. This is also suggested by the fact that the host range of coronaviruses is partly determined by the complementarity of the Spike receptor binding motif and the target hosts' ACE2 sequence [10,12]. A few studies [4,16–21] have addressed this question and gave rise to conflicting conclusions regarding the effects of specific variants on the interaction affinity and their relevance to the pandemic generally. The strongest of these used data from a published deep mutagenesis binding screen [22] to assess the effect of ACE2 population variants on Spike affinity and confirmed the effects of five key variants with further biochemical assays [17]. In our previous work [23], we employed the mCSM-PPI2 protein-protein interaction affinity prediction algorithm [13] to assess the effects of ACE2 variants on the binding of SARS-CoV-2 Spike and predicted that three reported ACE2 variants would strongly inhibit or abolish binding (p.Asp355Asn, p.Glu37Lys and p.Gly352Val). Confidence was given to these predictions by the performance of mCSM-PPI2[13] in comparison to binding data for 26 ACE2 mutants in complex with SARS-CoV Spike RBD [12]. Here we report experimental binding affinities of 10 ACE2 variants for isolated SARS-CoV-2 Spike RBD determined via surface plasmon resonance (SPR) at 37˚C. These results give better insight into the effect of ACE2 variants on SARS-CoV-2 Spike binding, revealing additional variants that enhance Spike binding, and also test the quality of our predictions. The SPR data allowed us to recalibrate the mCSM-PPI2 predictions to provide more accurate estimates of the effect of interface variants that we did not test experimentally.

## 2. Results and discussion

### 2.1. ACE2 variant affinities for SARS-CoV-2 Spike

Fig 1 highlights the mutated residues on the structure of ACE2 [11] in complex with SARS-CoV-2 Spike. We determined the binding affinity of 10 ACE2 mutants for isolated SARS-CoV-2 Spike RBD via surface plasmon resonance (SPR) to identify variants that may influence an individual's response to infection. Nine of these mutants were selected from the 241 ACE2 missense variants reported in gnomAD [24] on the basis of our previous computational predictions [23] and their reported prevalence, whilst the tenth mutant (p.Thr27Arg) was predicted to enhance Spike binding more than any other possible mutation at the interface.

Table 1 presents experimentally determined ΔΔG and mCSM-PPI2 [13] predictions for the 10 ACE2 mutants together with predicted data for a further three variants, alongside variant population frequencies and RBD interacting residues. Our SPR data were collected in two batches. The first batch comprised four variants, two were predicted to strongly reduce or abolish Spike binding (p.Glu37Lys and p.Asp355Asn) and two predicted to enhance Spike binding (p.Gly326Glu and Thr27Arg) [23]. The SPR measurements showed strongly reduced binding for p.Glu37Lys and the total abolition of binding for p.Asp355Asn (within the concentration range assayed), in agreement with the predictions. In contrast, p.Gly326Glu and Thr27Arg, which had been predicted to enhance binding, displayed decreased and slightly decreased binding, in disagreement with the predictions. These discrepancies motivated a second set of SPR measurements that included six of the most prevalent ACE2 variants close to the Spike binding site. Surprisingly, the two most common ACE2 variants tested bound SARS-CoV-2 Spike more strongly than reference ACE2. These were p.Lys26Arg

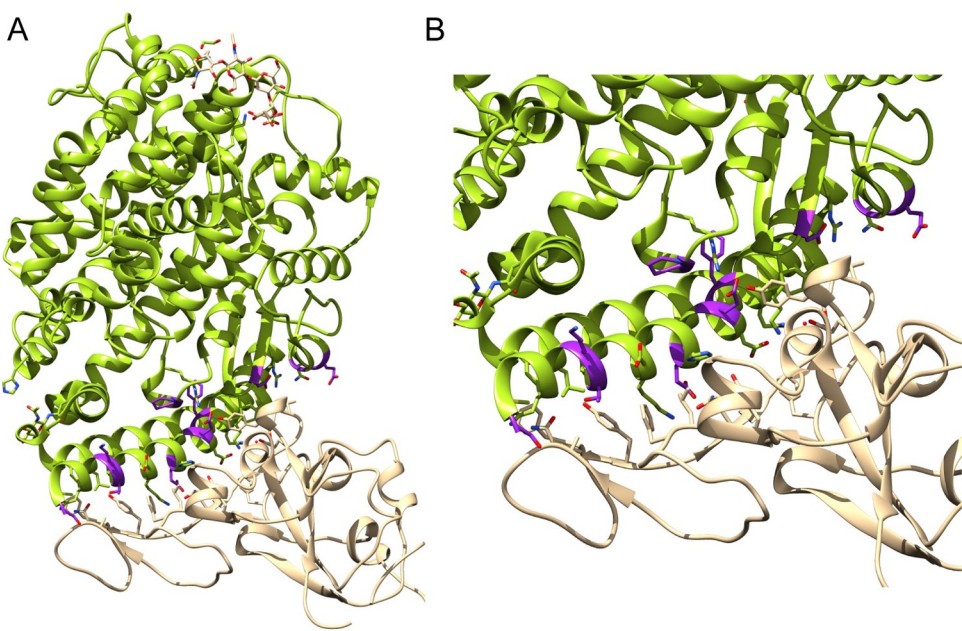

**Fig 1. Binding affinity determination of ACE2 variants with SARS-CoV-2 Spike.** A. ACE2 (green) in complex with Spike RBD (tan) from biological assembly 1 derived from PDB ID: 6vw1 [11]. The positions that were mutated in this work are highlighted magenta. B. The ACE2 Spike interface. Figure generated with Jalview [25] and UCSF Chimera [26].

($\Delta\Delta G$ = 0.26 ± 0.09), which has the highest allele frequency of ACE2 variants near the Spike interface, and p.Ser19Pro ($\Delta\Delta G$ = 0.59 ± 0.09) that has the second highest frequency. Two other variants in this series also increased Spike binding (p.Phe40Leu and p.Pro389His) despite being over 8 Å from the closest Spike residue. Finally, p.Glu329Gly may cause a slight reduction in binding ($\Delta\Delta G$ = -0.09 ± 0.09) whilst p.Glu35Lys had an inhibitory effect ($\Delta\Delta G$ = -0.36 ± 0.09). These results show that ACE2 variants can both enhance and inhibit Spike binding, properties that may reasonably be associated with susceptibility and resistance phenotypes to SARS-CoV-2 infection.

## 2.2. Structural features of affinity modifying variants

Figs 2 and 3 illustrate the mCSM-PPI2 provided models of p.Ser19Pro and p.Lys26Arg. p.Lys26Arg was not predicted to make any new well-defined contacts with Spike residues but it is predicted to adopt a conformation that extends toward Spike residues Y473 and F456, coming within 7.8 Å and 6.1 Å of these residues' aromatic rings, respectively, slightly beyond typical amino-aromatic interaction distances [27] but potentially favourable when dynamics are considered. Similarly, p.Ser19Pro also does not introduce any new Spike contacts according to the mCSM-PPI2 model and so the enhanced affinity is difficult to explain but it is has been suggested that the Pro mutant stabilises the helix to favour Spike interaction [22]. ACE2 p.Ser19Pro is of further interest because of its proximity to Spike Ser477, which has mutated to Asn in circulating SARS-CoV-2 strains and these ACE2 and RBD variants have been found to interact [28]. The mCSM-PPI2 structural models of the p.Asp355Asn, p.Glu37Lys and p.Gly352Val were discussed in detail in our previous work [23]. In summary, p.Asp355Asn was predicted to introduce a number of steric clashes at the interface, whilst p.Glu37Lys abolished an H-bond between ACE2 Glu37 and RBD Tyr505.

**Table 1. Surface plasmon resonance derived ΔΔG and mCSM-PPI2 [13] predictions for ACE2 mutants, including gnomAD [24] missense variants, at or near the ACE2 Spike interface.** See S1 Table for mCSM-PPI2 predictions for all gnomAD variants in the ACE2 ectodomain.

| Mutation | Distance to Spike | Spike residues | mCSM-PPI2 ΔΔG | Recalibrated mCSM-PPI2 ΔΔG | SPR ΔΔG | Max. prevalence (1 sf) |
|---|---|---|---|---|---|---|
| p.Ser19Pro | 2.6 | A475,G476,S477 | −0.2 | 0.2 | 0.59 ± 0.09 | 0.003 (AFR) |
| p.Lys26Arg | 6.0 | (F456) | 0.0 | 0.4 | 0.26 ± 0.09 | 0.006 (NFE) |
| p.Thr27Ala | 3.7 | A475,N487,F456,Y473,Y489 | −0.6 | −0.4 | - | 0.00007 (AMR) |
| p.Thr27Arg[a] | 3.7 | "" | 1.4 | [b](2.6) | −0.11 ± 0.10 | NA |
| p.Glu35Lys | 2.9 | Q493 | −0.5 | −0.3 | −0.36 ± 0.09 | 0.0001 (EAS) |
| p.Glu37Lys | 3.2 | Y505 | −1.2 | −1.3 | −1.33 ± 0.18 | 0.0003 (FIN) |
| p.Phe40Leu | 8.6 | (Y449,Q498) | -0.3 | 0.0 | 0.11 ± 0.09 | [c]0.0001 (AMR) [d]0.0002 (AFR) |
| p.Met82Ile | 3.5 | F486 | −0.3 | 0.1 | - | 0.0003 (AFR) |
| p.Gly326Glu | 5.5 | (V503,N506) | 1.0 | [b](2.1) | −0.65 ± 0.14 | 0.0001 (AFR) |
| p.Glu329Gly | 4.1 | R439 | −0.4 | −0.1 | −0.09 ± 0.09 | 0.0001 (NFE) |
| p.Gly352Val | 5.4 | (Y505) | −1.1 | −1.2 | - | 0.00005 (NFE) |
| p.Asp355Asn | 3.5 | G502,T500 | −1.3 | −1.5 | (< −3.16 ± 0.14)[e] | 0.00003 (NFE) |
| p.Pro389His | 8.1 | (Y505) | -0.1 | 0.4 | 0.27 ± 0.09 | 0.0002 (AMR) |

Column legend–Distance to Spike: The minimum distance of the wild-type residue to the SARS-CoV-2 Spike as resolved in PDB 6vw1 [11]. Spike residues: Spike residues within 5 angstrom of mutant site (or closest if no residues are in this range). mCSM-PPI2 ΔΔG: The predicted ΔΔG in kcal mol$^{-1}$ for the missense mutation calculated by mCSM-PPI2 [13] with PDB 6vw1 as the model structure. Recalibrated mCSM-PPI2 ΔΔG: adjusted mCSM-PPI2 ΔΔG following recalibration with SPR data (§2.4). SPR ΔΔG: ΔΔG determined by SPR assay. Max. prevalence: The allele frequency for the gnomAD continental population with the highest frequency (See the "Prevalence of ACE2-Spike affinity genotypes" section for all population frequencies and definitions of all gnomAD cohort abbreviations).

a. p.Thr27Arg was not reported in gnomAD and was selected as it had the highest predicted increased ΔΔG of any possible ACE2 mutation at the Spike interface [23].

b. These recalibrated predictions are extrapolations significantly beyond the affinity range used to calculate the recalibration curve. We also know that high predicted ΔΔG can be problematic and, in these two variants, are wildly incorrect.

c. p.Phe40Leu G>T allele.

d. p.Phe40Leu G>C allele.

e. No binding was observed for ACE2 p.Asp355Asn, this value corresponds to the calculated maximum affinity that is consistent with this observation.

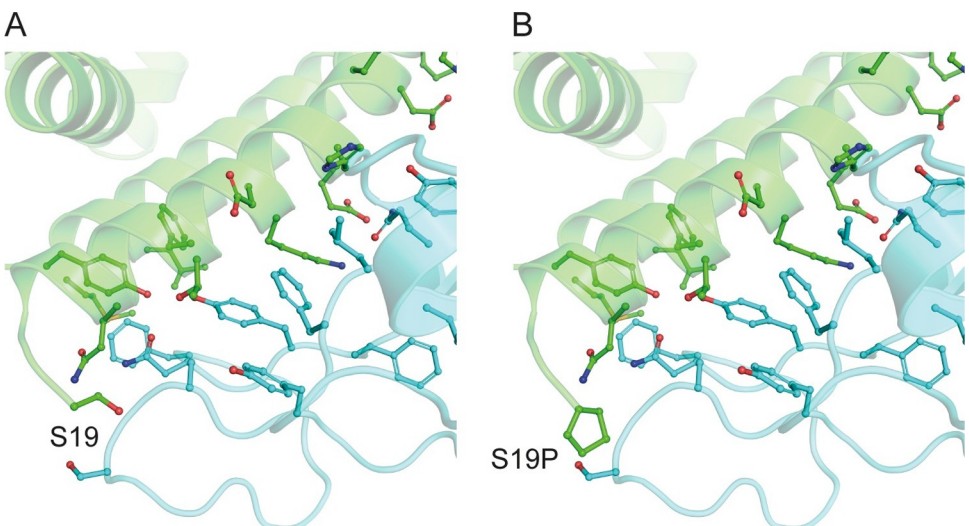

**Fig 2. The structure of ACE2 (green) gnomAD [24] missense variant p.Ser19Pro that enhances Spike (light blue) binding affinity.** A. The environment of ACE2 Ser19 from PDB ID: 6vw1 [11]. B. Model of ACE2 p.Ser19Pro in complex with Spike. The mutant structure was modelled onto 6vw1 with mCSM-PPI [13]. Figure created with PyMol [29].

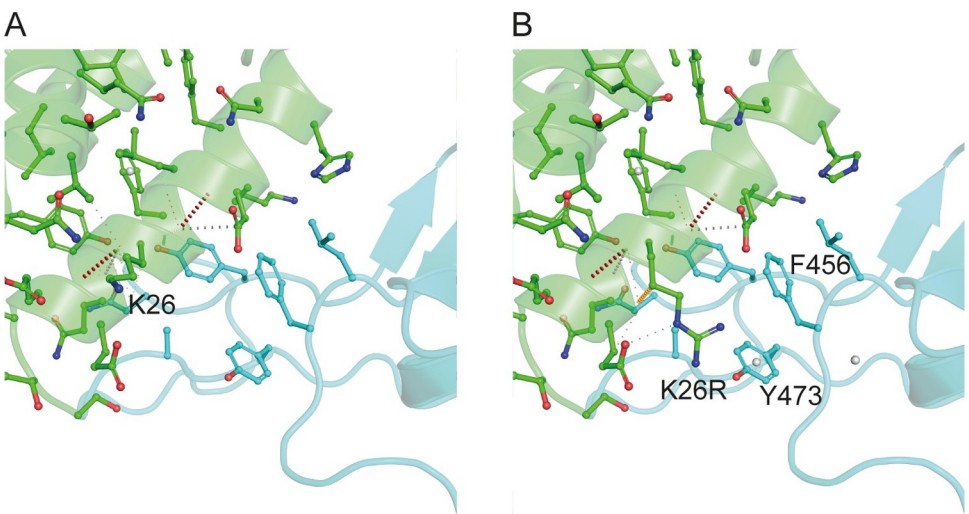

**Fig 3. The structure of ACE2 (green) gnomAD [24] missense variant p.Lys26Arg that enhances Spike (light blue) binding affinity.** A. The environment of ACE2 Lys26 from PDB ID: 6vw1 [11]. B. Model of ACE2 p.Lys26Arg in complex with Spike. The mutant structure was modelled onto 6vw1 with mCSM-PPI [13]. Figure created with PyMol [29].

## 2.3. Enhanced Spike binding ACE2 variants are relatively common in European (p.Lys26Arg) and African/African-American (p.Ser19Pro) populations

The enhanced binding by p.Lys26Arg and p.Ser19Pro is particularly interesting since this may effect carrier susceptibility or vulnerability toward SARS-CoV-2 infection and they have relatively high prevalence in the gnomAD [24] populations. p.Lys26Arg is the most common missense variant in the ACE2 ectodomain (Total allele count = 797, Total allele frequency = 0.004) and is predominant in the Ashkenazi Jewish cohort (ASJ AF = 0.01) and the European (non-Finnish) population (NFE, AF = 0.006). Amongst NFE sub-populations, it is most prevalent in north-western Europeans (AF = 0.007) and least prevalent in southern Europeans (AF = 0.003) and Estonians (AF = 0.003). p.Lys26Arg was also observed within this frequency range in Latino/Admixed American (AMR, AF = 0.003) samples. The variant is less frequent in Finnish (FIN, AF = 0.0005), African/African-American (AFR, AF = 0.001), South Asian (0.001) and, especially, East Asian (0.00001) samples. Interestingly, gnomAD reports a second variant at this site, p.Lys26Glu, suggesting that the position is especially tolerant to mutation. p.Ser19Pro is the next most common ACE2 missense variant in proximity to the Spike binding site (AC = 64, AF = 0.0003) and has the highest positive ΔΔG of all those tested (ΔΔG = 0.59 ± 0.03). This variant is practically unique to the African/African-American gnomAD population (AFR, AC = 63, AF = 0.003). The only other observation of this variant in gnomAD is in a single heterozygote in the labelled "Other" cohort. As a result of these distinct population distributions, it is possible that these variants could contribute to some of the observed epidemiological [30] variation between populations and ethnic groups.

Given the prevalence of these variants we checked for their occurrence in recent GWAS studies on Covid-19 related phenotypes. Table 2 presents GWAS association results for ACE2 p.Lys26Arg from the Covid-19 Host Genetics Initiative (HGI) [31]. These data show some consistency with the hypothesis that p.Lys26Arg contributes additional risk for more severe Covid outcomes, whilst not affecting the likelihood of infection. In the very severe respiratory confirmed Covid vs. population contrast (study A2), a non-statistically significant increased

**Table 2. Covid Host Genetics Initiative [31] Release 3 (accessed: 12th October 2020) summaries for p.Lys26Arg (rs4646116).**

| Covid HGI study and description | Beta | SE | p | p-het |
|---|---|---|---|---|
| A2—very severe respiratory confirmed covid vs. population | 0.38 | 0.24 | 0.12 | 0.74 |
| C1—covid vs. lab/self-reported negative | -0.18 | 0.14 | 0.21 | 0.92 |
| C2—covid vs. population | -0.03 | 0.11 | 0.81 | 0.65 |
| D1—predicted covid from self-reported symptoms vs. predicted or self-reported non-covid | -0.10 | 0.24 | 0.67 | 0.08 |

Column legend–Beta: average effect size of p.Lys26Arg on the phenotype described in the study description across all of the GWAS studies included in the Covid-HGI meta-analysis at this locus. SE: standard error of Beta. p: unadjusted p-value indicating the significance of association with the phenotype. p-het: between study heterogeneity p-value, which indicates consistency across studies.

risk was reported ($\beta = 0.38 \pm 0.24$, p = 0.12, p-het = 0.74; see Table 2 for definitions of these parameters). Other Covid HGI data summaries for contrasts testing for alleles associated with the risk of SARS-CoV-2 infection (i.e., studies C1, C2 and D1) suggest the variant does not play a role in infection acquisition. Although none of these tests achieved genome wide significance in the meta-analysis, it would be worthwhile to reassess this variant after controlling for other loci with the greatest effect sizes, sex, or other appropriate stratifications.

## 2.4. Recalibrated mCSM-PPI2 ΔΔG provides improved affinity predictions

Our SPR data provide accurate readouts of the effect of ACE2 variants on Spike binding, but we could not feasibly carry out experiments for all possible ACE2 mutations at the interface. For variants we have not studied, predictions and other high-throughput datasets can be useful, so long as their applicability and limitations are well-understood. In our previous work [23], we calibrated predictions against relative binding data for ACE2/SARS-CoV Spike, we now improve this by recalibrating the mCSM-PPI2 predictions with our ACE2/SARS-CoV-2 Spike SPR dataset.

Fig 4 compares experimental and mCSM-PPI2 [13] predicted ΔΔG for ACE2 mutants binding to Spike RBD and indicates the presence of two major groups of mutations. Amongst the seven mutants with detectable binding predicted to lower RBD affinity, the experimental and predicted ΔΔG values are highly correlated ($R^2 = 0.91$, p = 0.0006), implying only a slight systematic error in the predictions that could be corrected with a linear transformation. In contrast, our SPR data showed that the predictions of greatly enhanced binding for p.Gly326Glu and p.Thr27Arg were inaccurate. The different accuracies of affinity lowering and affinity increasing predictions were not unexpected given our previous calibration study, which showed poorer performance for affinity enhancing variants [23], and can be rationalised by observing that the mCSM-PPI2 training dataset contained far fewer experimentally determined affinity enhancing variants than affinity lowering variants [13]. Structural considerations may further account for the poor prediction obtained for p.Gly326Glu, because the additional RBD contacts predicted by mCSM-PPI2 for this mutant [23] could be offset by longer range consequences of the reduced torsional flexibility at mutant Glu326 and it is reasonable to expect that mCSM-PPI2 lends greater weight to local effects given that a key feature of the model is a graph representation of the residue contact network [13]. Accordingly, we argue that because we can account for the inaccuracy of the two high ΔΔG predictions in our dataset, it is reasonable to rescale mCSM-PPI2 predicted ΔΔG using the linear relationship between the seven well-correlated predictions (i.e., omitting p.Gly326Glu and p.Thr27Arg) to provide

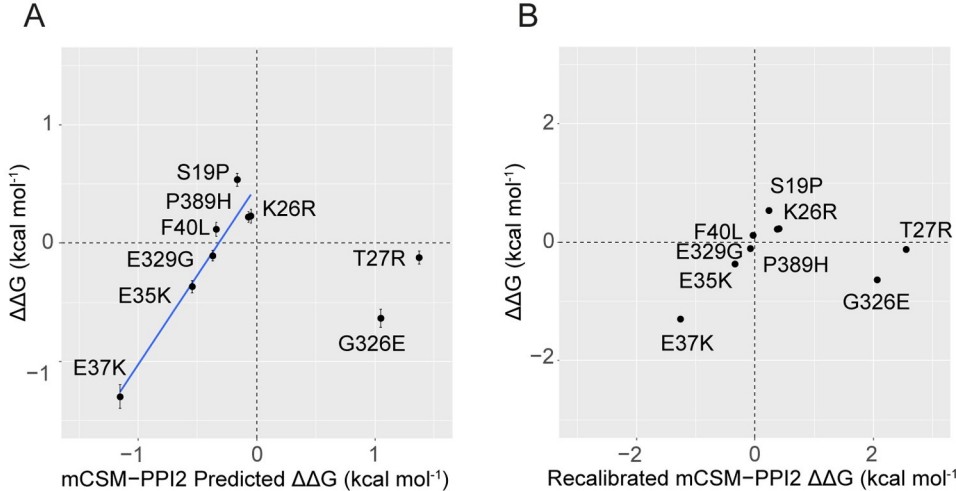

**Fig 4.** A. mCSM-PPI2 predicted ΔΔG is linearly correlated with SPR determined ACE2 variant Spike RBD affinities for ACE2 variants with negative predicted ΔΔG. The linear model shown was fit to the eight ACE2 mutants with predicted ΔΔG < 0 kcal mol$^{-1}$ (slope = 1.56, intercept = 0.52, R$^2$ = 0.91, p = 0.0006, n = 7). As explained in the text, p.Thr27Arg and p.Gly326Glu were excluded from the fit because the SPR data show that the mCSM-PPI2 prediction for these variants is incorrect, in keeping with the poorer performance of affinity predictions involving higher ΔΔG [23]. The recalibration is thus strictly applicable only to predicted ΔΔG within the range of interpolation. p.Asp355Asn is also not included because in our SPR experiments this ACE2 mutant could not be detected binding to RBD and so we do not have an exact experimental affinity for this mutant; inclusion of this variant's estimated maximum ΔΔG (−3.16 kcal mol$^{-1}$) as a proxy value, alters the fit but does not change our conclusions. B. Recalibrated mCSM-PPI2 predicted ΔΔG correctly classifies ACE2 variants that enhance RBD affinity. Figure generated with R ggplot2.

better predicted ΔΔGs; especially within the range of interpolation between p.Glu36Lys (ΔΔG$^{pred}$ = −1.2 kcal mol$^{-1}$) and p.Lys26Arg (ΔΔG$^{pred}$ = 0.0 kcal mol$^{-1}$). We therefore produced recalibrated predictions by transforming the mCSM-PPI2 prediction with the linear model obtained with regression (see Methods). Whilst this recalibration cannot correct the problem with the grossly inaccurate predictions for p.Gly326Glu and p.Thr27Arg, the recalibrated ΔΔG yields correct qualitative predictions for all eight mutations with negative predicted ΔΔG and measurable ΔΔG from SPR (Table 1). Notably, this lends additional confidence in our prediction that ACE2 p.Gly352Val strongly reduces Spike binding (in agreement with published deep mutagenesis experiments [22]).

Fig 5 compares predicted ΔΔG to the relative ACE2 mutant binding measurements from Procko and coworkers [22] who determined the binding of ACE2 mutants to Spike RBD relative to ACE2 WT by a deep mutagenesis protocol. In this approach, human Expi293F cells expressing an ACE2 mutant library were sorted into high and low RBD binding populations that were then sequenced to determine the proportions of ACE2 mutants in each set producing the enrichment scores plotted in Fig 5. These data provide an additional benchmark for the predicted binding affinities, especially to compare the original mCSM-PPI2 prediction to the recalibrated prediction. However, it is important to recognise that the deep mutagenesis assay provides relative binding scores that are influenced by variable ACE2 cell-surface expression amongst the mutants as well RBD affinity [22], and therefore these data should not be considered a gold standard for the affinities predicted in this work.

Despite this caveat, amongst the 113 mutations with recalibrated ΔΔG < −1 kcal mol$^{-1}$, there is excellent agreement with the deep mutagenesis assay where 108 show decreased binding in the deep mutagenesis assay (i.e., enrichment scores < 0; 96% agreement), which suggests that low predicted ΔΔG is a highly specific indicator of inhibited binding (n.b., the

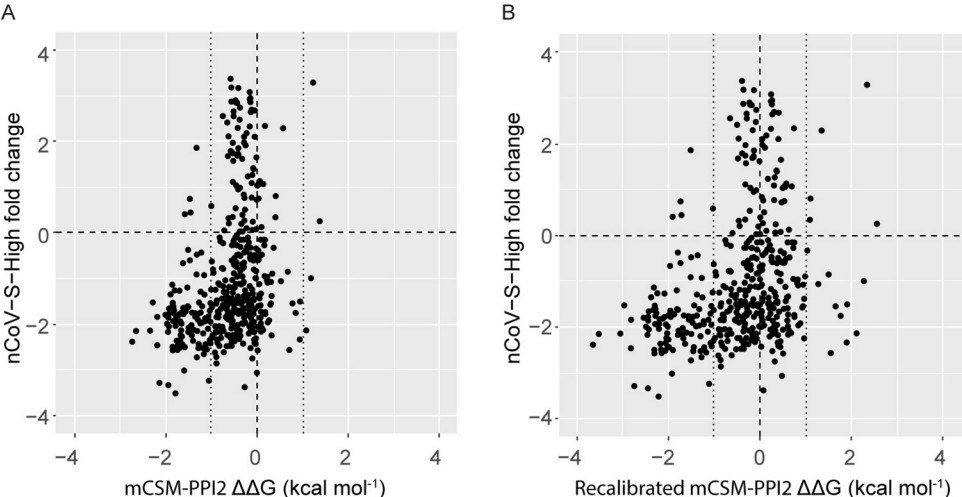

**Fig 5. mCSM-PPI2 predicted ΔΔG with and without recalibration are in agreement with published deep mutagenesis binding data [22] at low predicted ΔΔG but recalibrated predictions are more sensitive towards variants that increase binding.** ACE2 mutant binding to Spike RBD relative to ACE2 WT determined via a deep mutagenesis protocol vs. A. mCSM-PPI2 predicted ΔΔG and B. recalibrated predicted ΔΔG for all 437 possible ACE2 mutants at the 23 sites within 5 Å of Spike RBD. The dotted lines highlight predicted ΔΔG ±1 kcal mol$^{-1}$. The overwhelming majority of predictions by mCSM-PPI2 below this range also show reduced binding in the deep mutagenesis binding data, suggesting mCSM-PPI2 is specific for mutants that significantly lower binding, in agreement with our previous determination using a different dataset [23]. Within ±1 kcal mol$^{-1}$, the deep mutagenesis assay finds variants that increase and lower binding and whilst the original mCSM-PPI2 predictions predicts most of these to lower binding, the recalibrated predictions correctly reflect the ambiguity by centring these variants at predicted ΔΔG = 0 kcal mol$^{-1}$. Above this range (i.e., > 1 kcal mol$^{-1}$), where variants are predicted to enhance binding, a higher proportion of the recalibrated predictions also show enhanced binding in the deep mutagenesis data (5/17) compared to the raw mCSM-PPI2 prediction (2/5). Figure generated with R ggplot2.

original mCSM-PPI2 predictions behave similarly, where 105 out of 110 variants with predicted ΔΔG < −1 kcal mol$^{-1}$ show decreased binding, in 95% agreement). For the 324 ACE2 mutants with recalibrated ΔΔG > −1 kcal mol$^{-1}$, the deep mutagenesis assay identifies 81 mutations with increased binding and 243 with decreased binding. The original mCSM-PPI2 prediction yields ΔΔG < 0 kcal mol$^{-1}$ for most of these variants (i.e. 266 variants have ΔΔG between −1 and 0 kcal mol$^{-1}$ vs. 61 that have predicted ΔΔG > 0 kcal mol$^{-1}$; S4 Table), compared to a balanced set of negative and positive ΔΔG predictions after recalibration (163 vs. 161, respectively), which may better reflect the ambiguity of predictions in this range. These observations are reflected in the fact that classification of ACE2 mutants as affinity increasing or decreasing based on recalibrated ΔΔG is significantly associated with the deep mutagenesis binding data at a threshold of 0 kcal mol$^{-1}$ ($\chi^2$ = 10, p = 0.001; S3 and S4 Tables whilst the raw mCSM-PPI2 predictions are not ($\chi^2$ = 0.27, p = 0.6).

If we were to treat the deep mutagenesis assay as a gold standard for the affinity predictions (i.e., assume the effects of cell-surface expression are small), we may conclude that: 1) ΔΔG < −1 kcal mol$^{-1}$ is a highly specific indicator of significantly decreased affinity, 2) a predicted affinity in the range −0.5 ≤ ΔΔG < 0 kcal mol$^{-1}$ (or recalibrated −0.5 ≤ ΔΔG < 0.5 kcal mol$^{-1}$) is an ambiguous prediction and 3) recalibrated ΔΔG is more sensitive towards variants that increase affinity compared to the original prediction but remains prone to false predictions of increased affinity (S4 Table). However, if the effects of variable cell-surface expression were to account for a large proportion of the discrepancies between these datasets then alternative conclusions are that: 1) avidity effects rarely rescue binding when ΔΔG < −1 kcal mol$^{-1}$, whereas 2) cell-surface expression has a substantial effect on binding when the change in

affinity is small, and 3) decreased cell-surface expression can abrogate binding even when the affinity is very high. Since the contribution of each of these limiting cases will vary on a per variant basis, further experimental data are required to distinguish the relative importance of each scenario overall. However, comparing the deep mutagenesis binding data to our experimentally determined ΔΔG suggests that a combination of these effects is at play (S2 Fig).

## 2.5. Predicted burden of rare ACE2 variants with Spike affinity phenotypes

Existing human variation datasets are well-powered to detect common variation (1KG was estimated to detect >99% SNPs with MAF >1% [32] and gnomAD is substantially larger) in the sampled populations but they are far from comprehensive with respect to rare variation [24]. Rare variants in ACE2 that influence Spike binding could have implications for the epidemiology of COVID-19 in addition to the consequences for affected individuals. If there were 10 such variants with an allele frequency of 1 in 50,000, their collective occurrence might be as high as 1 in 5,000 (discounting linkage) and when this is considered alongside the possibility that a high proportion of the global population will be exposed to SARS-CoV-2 it becomes clear that such effects should be investigated. These variants could even be present at significant frequencies in populations missing or underrepresented in gnomAD.

Fig 6 illustrates the distribution of recalibrated mCSM-PPI2 ΔΔG predictions ($\Delta\Delta G^{recal}$) for all 475 possible ACE2 mutations at 25 ACE2 residues close to the Spike interface and the subset that are accessible via a single base change of the ACE2 coding sequence (these are more likely to be present in human populations than those requiring multiple substitutions). Most of these mutations are predicted to have only a slight effect on Spike binding, but there is a secondary mode below −1.0 kcal mol$^{-1}$ with 126 mutations predicted to lead to strongly reduced binding. Fewer variants received high positive $\Delta\Delta G^{recal}$ scores: 17 had $\Delta\Delta G^{recal}$ > 1.0 kcal mol$^{-1}$, a further 44 had $\Delta\Delta G^{recal}$ > 0.5 kcal mol$^{-1}$ and an additional 70 had $\Delta\Delta G^{recal}$ > 0.2 kcal mol$^{-1}$ (i.e., $\Delta\Delta G^{recal}$ > p.Lys26Arg). A similar pattern is observed for the 151 mutants corresponding to 172 single nucleotide variants of the ACE2 coding sequence (Fig 6B). These results suggest that random novel ACE2 missense variants at these loci can inhibit or enhance Spike binding, but are more likely to be inhibitory, meaning that diversity at these positions might typically

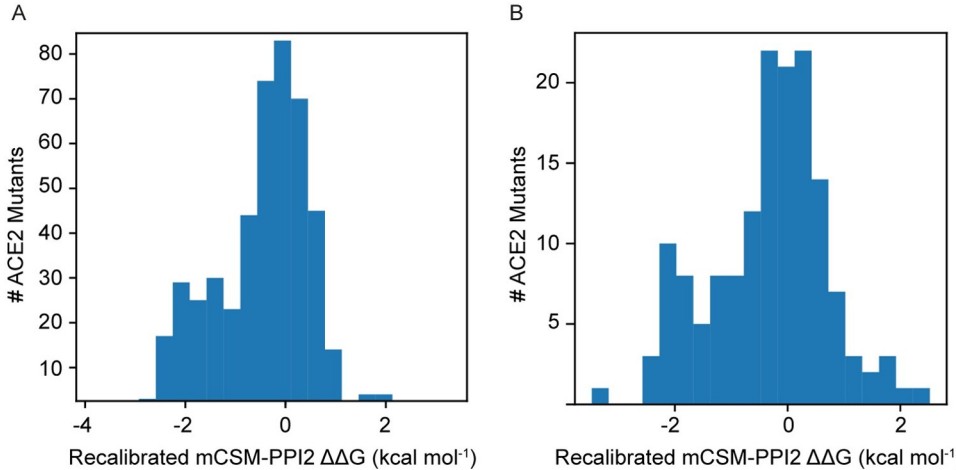

**Fig 6.** Distribution of mCSM-PPI2 [13] predicted ΔΔG from in silico saturation mutagenesis of the ACE2-S interface in PDB 6vw1 [11]. A. predicted ΔΔG for 475 mutations across 25 sites on ACE2 corresponding to the 23 residues within 5 Å of SARS-CoV-2 S plus Gly326 and Gly352. B. predicted ΔΔG for the subset of 151 mutations across these sites that are accessible via a single base change of the ACE2 coding sequence.

be beneficial and provide some resistance to infection. Notably, these recalibrated predictions do not display the same bias toward slightly negative $\Delta\Delta G$ for mutations predicted to cause small changes in binding that the raw mCSM-PPI2 $\Delta\Delta G$ predictions did [23], whilst still making more assertive predictions of mutations that disrupt binding (S1 Fig), which may be another indicator of the improved quality of the recalibrated predictions.

Even though most of these variants are not reported in gnomAD, they may still occur within the populations represented, especially if they occur at frequencies that are poorly detected. With this in mind, we calculated allele frequencies for these mutations that are compatible with their absence from gnomAD (see Methods) in order to gain a better idea of how widespread the effects of these variants might be.

Table 3 presents estimated allele frequencies for novel variants that are predicted to inhibit and enhance Spike binding at varying $\Delta\Delta G^{recal}$ thresholds. Upper bounds for their joint frequencies assuming that they occur at lower frequencies than singleton variants suggest frequency bounds that range 1 in 1,000 to 1 in 2,000 variants per allele for inhibitory variants, and 1 in 1,500 to around 1 in 5,000 for enhancer variants. A second approach, which takes account of the empirical detection of rare variants in ACE2, yields frequencies that span from 1 in 6,250 to 1 in 12,195 for inhibitory variants, and 1 in 8,333 to 1 in 37,037 for enhancer variants. These estimates show that novel variants in ACE2 with any weak Spike affinity phenotype could plausibly be as common as 1 in 3,571 alleles (calculated as the sum of the highest inhibitor and enhancer prevalence's), but the strongest affinity phenotypes are more likely to occur in frequency ranges akin to rare genetic diseases. It should be remembered that these values are calculated to be compatible with their absence from the gnomAD dataset, but it remains possible that some or all of these variants do not exist at all or that they are very common in one or more populations not represented in gnomAD.

## 2.6. Impact of ACE2 affinity variants on SARS-CoV-2 infection

How far can affinity modulating variants influence SARS-CoV-2 infection? For inhibitory variants, host range specificity and mutagenesis studies [12] highlight how some receptor mutations can provide complete protection from infection. We identified one variant (p.Asp355Asn) that abolished Spike binding altogether within our detection limits and a few others that reduced binding to varying extents. Since these variants were observed in population samples, some individuals carry ACE2 variants that could confer complete resistance to

**Table 3. Estimated allele frequencies of potential novel variants in ACE2 with predicted Spike binding phenotypes.** The estimate is calculated from the observed occurrence of rare variants in ACE2 in gnomAD [24] ($6.1 \times 10^{-6}$) and the proportion of SNPs at the 25 residues that are predicted to modify Spike affinity at different thresholds (see Methods).

| $\Delta\Delta G^{recal.}$ (kcal mol$^{-1}$) | N missense | N SNPs | Joint singleton frequency (NFE) | Joint estimated frequency (NFE) | Heterozygotes / 100K | Hemizygotes / 100K |
|---|---|---|---|---|---|---|
| | | | Predicted Spike inhibitory variants | | | |
| $< -1.0$ | 38 | 41 | 0.0005 | $8.2 \times 10^{-5}$ | 16.5 | 8.2 |
| $< -0.5$ | 55 | 59 | 0.0007 | 0.00012 | 23.8 | 11.9 |
| $< -0.2$ | 77 | 81 | 0.001 | 0.00016 | 33.0 | 16.5 |
| | | | Predicted Spike enhancer variants | | | |
| $> 1.0^{a,b}$ | 11 | 14 | 0.0002 | $2.7 \times 10^{-5}$ | 5.5 | 2.7 |
| $> 0.5^{a}$ | 24 | 30 | 0.0004 | $6.0 \times 10^{-5}$ | 11.9 | 6.0 |
| $> 0.2$ | 48 | 58 | 0.0007 | 0.00012 | 23.8 | 11.9 |

a. These variants are outside the interpolation range of our experimental recalibration of mCSM-PPI2 predictions and are potentially less reliably predicted.

b. These variants have predicted $\Delta\Delta G$ akin to the inaccurately predicted p.Thr27Arg and p.Gly326Glu variants and are potentially the least reliably classified.

infection. Variants that reduce but do not eliminate binding may confer a degree of resistance proportional to the affinity reduction, or alternatively, there may be an affinity threshold that toggles cellular permissivity as in other enveloped viruses [33].

It is less clear what effect affinity enhancing variants have on virulence, but there are indications that in some cases carriers may be at greater risk of infection and severe disease. Virus attachment proteins in enveloped viruses require a minimum receptor affinity that is proportional to the receptor surface density on the target cell to enable membrane fusion [33]. If this applies to SARS-CoV-2, ACE2 variants that enhance Spike binding could increase viral spreading in carriers due to increased cellular tropism. Greater viral spreading is associated with clinical deterioration [34] and could cause increased infectiousness, akin to the enhanced transmissibility of SARS-CoV-2 vs. SARS-CoV, which is associated with increased viral loads in the upper respiratory tract [6,35] and also correlates with enhanced receptor affinity [11]. Similarly, the Spike N501Y mutation causes greatly enhanced ACE2 affinity [28] and is also associated with increased transmissibility [36]. Indeed, the recent report that Spike N501Y increases viral replication fitness in the upper airway of male golden Syrian hamsters (and Human airway epithelial cells), whilst showing only slightly increased fitness in lung tissues is also consistent with this idea [37].

A few studies have produced infectivity data for SARS-CoV-2 towards cells expressing ACE2 variants that allow us to compare ACE2 variant specific affinities and infectivity directly. Early work with SARS-CoV-2 pseudotypes showed slightly decreased infectivity towards cells expressing ACE2 p.Lys26Arg and no difference in the susceptibility of cells expressing ACE2 p.Ser19Pro [16]. In more recent work Shukla et al. [38] reported that the ACE2 mutants p.Lys31Asp and p.Lys353Asp caused only minimal changes in infectivity despite having substantially reduced RBD affinity. This surprising result prompted the authors to examine the interplay between ACE2 cell-surface expression and RBD affinity and they discovered that the high ACE2 surface expression in their initial assay masked the effects of impaired binding on infectivity; when cells were modified to express lower levels of ACE2 (which in itself decreased susceptibility) the two mutants displayed substantially reduced infectivities towards SARS-CoV-2 pseudotypes compared to ACE2 WT [38]. The authors also highlighted that 53% of tissues and cell-types within the GTEx database expressed ACE2 at levels that were possibly comparable to those in the Kozak modified cells that displayed RBD affinity dependent SARS-CoV-2 entry [38]. This result lends further support to the idea that ACE2 affinity variants could influence cellular tropism.

Shukla et al. [38] proceeded to determine SARS-CoV-2 pseudotype infectivities towards 28 ACE2 mutants and Fig 7 shows how these compare to our experimental and predicted ACE2 and Spike RBD affinity data. Amongst the six ACE2 variants with SPR and infectivity data, all variants that decreased affinity also decreased infectivity whilst the two that enhanced affinity also showed increased infectivity (although the increase is marginal for p.Lys26Arg). Moreover, there is a remarkable quantitative association between RBD affinity and pseudotype infectivity that fits well to a negative exponential relationship (Fig 7A; $p = 0.05$). In this kind of relationship, there is a ceiling to the increase of infectivity due to increasing affinity as the curve is bounded above. This is concordant with the affinity-infectivity relationships observed in other enveloped viruses [33], and anticipates the behaviour of Spike variant p.Asn501Tyr ("N501Y") for which Shukla et al. [38] reported almost no change in SARS-CoV-2 pseudotype infectivity towards ACE2 WT despite a near 10-fold increase in RBD affinity ($\Delta\Delta G = 1.43$ kcal mol$^{-1}$) [28]. These observations support the hypothesis that affinity lowering mutations can be expected to protect the host from infection to an extent proportional to the reduced affinity, but they do not completely clarify the effect of affinity enhancing mutations, showing only that

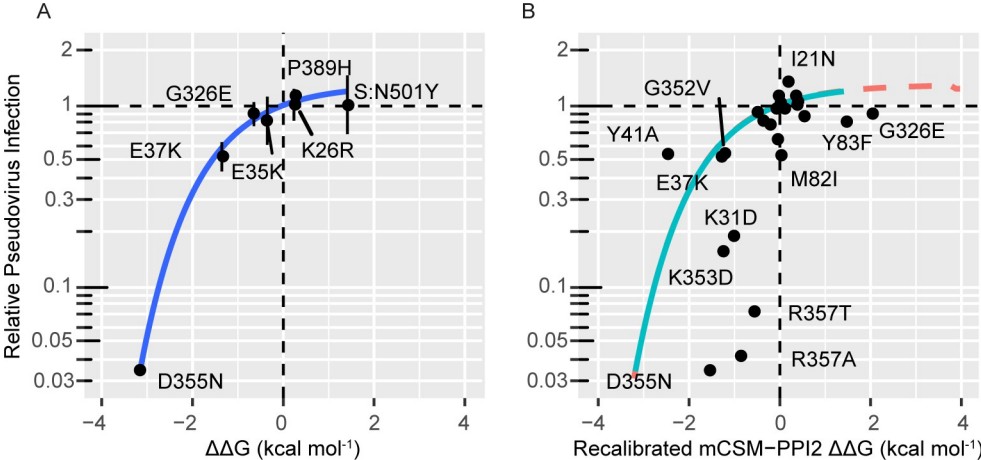

**Fig 7. The relationship between experimental and predicted ΔΔG of ACE2 mutants and relative SARS-CoV-2 pseudotype infectivity reported by Shukla et al [38].** A. Pseudotype infectivity is strongly correlated with experimental ΔΔG and fits well to a negative exponential model ($y = a(1−e^{−cx})$; a = 0.25, p = 0.05; c = 0.84, p = 0.0007), which is in line with the mechanism proposed by Hasegawa et al. for other enveloped viruses [33], where there is an cell specific affinity threshold for infection beyond which infectivity does not increase. To extend the dataset we have included the affinity [28] and infectivity [38] data for the Spike N501Y variant. n.b. ACE2 p.Asp355Asn did not bind Spike RBD in our experiments and we have imputed this result with the calculated upper bound based on the detection limits of the assay. B. Pseudotype infectivity is also associated with recalibrated mCSM-PPI2 predicted ΔΔG (Spearman ρ = 0.69, p = 0.0003), this is a stronger association than reported by Shukla et al. [38] who compared infectivity to Procko and co-workers [22] deep mutagenesis binding data (Spearman ρ = 0.51). The model derived from the regression of pseudotype infectivity against experimental ΔΔG is overlayed for comparison; the region that is extrapolated beyond the range of the experimental ΔΔG data is indicated by the red dashes.

increased affinity variants can be associated with minor increases in cellular susceptibility but can also have no effect on this property.

Predicted ΔΔG is also significantly correlated with pseudotype infectivity (Fig 7B; Spearman ρ = 0.69, p = 0.0003, n = 24), and there is a suggestive association between a predicted decrease in affinity and decreased infectivity ($\chi^2$ = 3.66, p = 0.06). This is a stronger association than reported by Shukla *et al.* [38] who compared infectivity to Procko and co-workers [22] deep mutagenesis binding data, reporting Spearman ρ = 0.51. Notably all 6 ACE2 mutants with pseudotype infectivity greater than ACE2 WT had positive predicted ΔΔG whilst most of those with significantly reduced susceptibility had ΔΔG < −0.5 kcal mol⁻¹. This suggests that a low predicted ΔΔG may imply reduced infectivity whilst a positive predicted ΔΔG is sometimes associated with increased infectivity. Although error in the predicted affinities could account for part of the discrepancy, some of the ACE2 mutants may vary in other parameters besides affinity. For example, ACE2 R357A was shown to have decreased ACE2 expression [38] and this no doubt contributes to this mutation's decreased infectivity in synergy with the reduced RBD affinity of this mutant. Shukla et al. attempted to characterise the relationship between infectivity and affinity by comparing their pseudotype infectivity data to the deep mutagenesis binding assay reported by Procko and co-workers [22], but this comparison also revealed a non-linear relationship with several exceptions and discrepancies [38]. Even when considering only the six variants in Fig 7A, the deep mutagenesis binding results do not provide as striking a correlation as seen here. Our success is probably in part due to the highly accurate affinities provided by the SPR assay, alongside the serendipitous occurrence that the six ACE2 variants for which we could perform this comparison may differ only in affinity and no other key parameters (e.g., receptor surface expression levels).

Other considerations relevant to the effect of ACE2 Spike affinity variants on carriers with respect to SARS-CoV-2 infection include ACE2 carboxypeptidase activity [39], which may be modulated by the specific binding affinity, the effect of hemizygosity and sex differences in Covid-19 outcomes, and the interplay of affinity variants and ACE2 expression levels. The importance of ACE2 expression was mentioned earlier but it is worth highlighting that it is known to partly determine the cellular specificity of SARS-CoV [40] and was explored as a potential factor in COVID-19 susceptibility and severity, including the interaction between ACE2 variants and ACE2 stimulating drugs [41]. Also, heterozygotes express a proportion of ACE2 alleles whilst hemizygotes carry only a single ACE2 allele so that ACE2 Spike affinity variants ought to always show greater penetrance in hemizygotes, for better or worse. In contrast and since ACE2 escapes complete X-inactivation [20], heterozygotes have the advantage that the more resistant ACE2 allele could become dominant in infected tissues due to selection over cellular infection cycles, gradually increasing the prevalence of the more Spike resistant ACE2 allele. Besides the possible benefit against infection, there could be other implications for heterozygotes depending on the persistence of X-inactivation bias and the nature of any hitchhiking alleles.

A further effect of ACE2 variants on SARS-CoV-2 infection is the potential interaction between Spike RBD variants and ACE2 variants. ACE2 variants may interact with Spike variants either additively or non-additively depending on their structural complementarity, which could give rise to differential effects of SARS-CoV-2 Spike variants towards individuals that carry ACE2 variant alleles. For instance, in other work where we focussed on the effects of the RBD variants in the Alpha and Beta strains, we identified an antagonism between ACE2 p.Ser19Pro and Spike S477N whereas Spike mutations from the Alpha and Beta strains had a largely additive effect on affinity with the ACE2 p.Ser19Pro or p.Lys26Arg variants [28]. Another example is that the Spike N501Y variant rescues infectivity towards ACE2 variants p.Lys355Asp, p.Asp355Asn and p.Asp38His [38], which presumably is in part due to compensation of the reduced affinity of these ACE2 variants by the increased affinity of Spike N501Y. Interestingly, the infectivity of ACE2 p.Arg357Ala, p.Glu37Lys and p.Tyr41Ala were not increased by Spike N501Y [38] even though the greatly enhanced affinity of this mutation towards ACE2 WT ($\Delta\Delta G$ = 1.43 kcal mol$^{-1}$) [28] should significantly compensate for these variants. As previously noted, ACE2 p.Arg357Ala displayed significantly reduced expression compared to ACE2 WT[38], which accounts for the insensitivity of this variant towards Spike N501Y. However, p.Glu37Lys and p.Tyr41Ala showed expression comparable to ACE2 WT (although the data for p.Glu37Lys were particularly variable) [38] and so these observations suggest an antagonistic relationship between the affinities of these ACE2 variants and Spike N501Y.

## 2.7. Prevalence of ACE2-Spike affinity genotypes

Table 4 presents detailed allele frequencies from gnomAD [24] for the ACE2 variants investigated in this work. The most prevalent ACE2 variant mediated Covid-19 phenotypes are likely to arise from the two relatively common variants found to enhance Spike binding, p.Ser19Pro and p.Lys26Arg. These variants were found in 3 in 1,000 individuals in the gnomAD African/African-American (AFR) samples and 7 in 1,000 North Western non-Finnish European samples (NW-NFE), respectively. p.Lys26Arg was also observed in other populations, including 1 in 1,000 AFR samples, adding further burden to this cohort. The high affinity displayed by ACE2 p.Ser19Pro is concerning in context with the reported disproportionate impact of Covid-19 on black and minority ethnic groups[30]. Further research to identify the impact of these variants on SARS-CoV-2 pathogenesis should be prioritised with urgency and existing

**Table 4. Detailed prevalence data from gnomAD [24] for ACE2 variants reported in Table 1.**

| Variant | ΔΔG (kcal mol⁻¹) | AFR | AMR | ASJ | EAS | FIN | NFE | SAS | OTH |
|---|---|---|---|---|---|---|---|---|---|
| p.Ser19Pro | 0.59 | 0.003 | - | - | - | - | - | - | 0.0002 |
| p.Lys26Arg | 0.31 | 0.001 | 0.003 | 0.01 | 0.00007 | 0.0005 | 0.006 | 0.001 | 0.003 |
| p.Thr27Ala | (−0.4) | - | 0.00007 | - | - | - | - | - | - |
| p.Glu35Lys | −0.29 | - | - | - | 0.0001 | - | 0.00001 | - | - |
| p.Glu37Lys | −1.25 | 0.0001 | - | - | - | 0.0003 | - | - | - |
| p.Phe40Leu | 0.14 | [a]0.0002 | [b]0.0001 | - | - | - | - | - | - |
| p.Met82Ile | (0.1) | 0.0003 | - | - | - | - | - | - | - |
| p.Gly326Glu | −0.67 | 0.0001 | - | - | - | - | - | - | - |
| p.Glu329Gly | −0.06 | - | - | - | - | - | 0.00007 | - | 0.0002 |
| p.Gly352Val | (−1.2) | - | - | - | - | - | 0.00001 | - | - |
| p.Asp355Asn | No binding | - | - | - | - | - | 0.00003 | - | - |
| p.Pro389His | 0.12 | - | 0.0002 | - | - | - | 0.00002 | - | - |
| Total frequency | | 0.0047 | 0.00317 | 0.01 | 0.00017 | 0.0008 | 0.00614 | 0.001 | 0.0034 |
| Enhancers (≥0.1 kcal mol⁻¹) | | 0.0045 | 0.0031 | 0.01 | 0.00007 | 0.0005 | 0.0062 | 0.001 | 0.0032 |
| Inhibitors (≤ −0.1 kcal mol⁻¹) | | 0.0002 | 0.00007 | - | 0.0001 | 0.0003 | 0.00005 | - | - |

AFR: African/African-American, AMR: Latino/Admixed American, ASJ: Ashkenazi Jewish, EAS: East Asian, FIN: Finnish, NFE: non-Finnish European, SAS: South Asian, OTH: Other

and future GWAS should investigate these variants more closely with appropriate stratifications.

The remaining gnomAD variants predicted to effect Spike binding are all very rare. The two other variants in our SPR series that also enhanced Spike binding have a joint frequency of around 3 in 10,000 in Latino/Admixed American (AMR) samples whilst amongst Spike inhibitory variants, p.Glu37Lys is the most prevalent occurring in 3 in 10,000 in Finnish samples and one additional African/African American sample. The other strongly inhibitory variants p.Gly352Val (predicted) and p.Asp355Asn are both doubletons observed only in non-Finnish Europeans, corresponding to an extremely low allele frequency of 3 in 100,000. Notably, the inhibitory variant p.Glu35Lys is the only variant predominant in East Asian samples.

Our analysis of the effects of all possible missense mutations at the ACE2 Spike interface predicts many other potential novel variants that effect the Spike interaction, but even though a relatively high proportion of possible variants are predicted to effect Spike binding (up to 62%), their collective frequency could be as little as 2.8 in 10,000. These frequencies are comparable to those that cause rare diseases. However, even though we find that ACE2 alleles with large positive or negative ΔΔG are likely to be extremely rare, it is important to note that allele frequencies show significant variation even between large variation datasets [42]. Moreover, it is known that local population allele frequencies can vary substantially from those reported in public datasets [43] and it is therefore possible that some populations have risk or protective mutations at higher frequencies. It is also worth highlighting that the substantially increased Spike affinities of the two most common variants tested may indicate the presence of a past selective effect occurring twice independently, which may increase the possibility of other higher frequency variants amongst populations not well represented by gnomAD.

## 3. Conclusion

One of the initial steps of SARS-CoV-2 infection is the attachment of the Spike protein to the ACE2 receptor on target cells and it is an open question whether individuals with missense

variants in this receptor could have different susceptibilities towards SARS-CoV-2 infection or severe COVID-19. We determined the binding affinities of 10 ACE2 mutants, including 9 variants reported in gnomAD [24], towards SARS-CoV-2 Spike RBD with a carefully designed SPR assay to obtain accurate and physiologically relevant measurements. We found that ACE2 p.Ser19Pro and p.Lys26Arg, two of the most common ACE2 missense variants in gnomAD [24], possessed substantially increased affinity for RBD, which raises the possibility that they could be risk factors for severe COVID-19. These variants have distinct distributions across the gnomAD cohorts and so any phenotypic effects they have could manifest population scale differences. Two other variants (p.Pro389His and p.Phe40Leu) were identified that also enhanced Spike binding, but these were relatively rare in the gnomAD cohorts. ACE2 variants also reduced RBD affinity, including p.Glu37Lys and p.Asp355Asn, which strongly inhibited and abolished binding, respectively, and could therefore protect carriers against infection or severe COVID-19. Amongst the other gnomAD variants tested, two variants inhibited binding (p.Glu35Lys and p.Gly326Glu) and the other did not show a significant difference in binding (p.Glu329Gly).

To include ACE2 variants in our analysis that we did not assess experimentally, the SPR affinity data were used to recalibrate mCSM-PPI2 [13] predicted ΔΔG to provide improved predictions for all possible ACE2 missense variants that interact with Spike. The validity of recalibration was confirmed by comparing the ΔΔG predictions to published binding data determined via deep mutagenesis [22], and subsequently we estimated the prevalence of novel Spike affinity modifying variants in ACE2. A key feature of our burden assessments was to distinguish between variants predicted to inhibit and enhance Spike affinity, since these are expected to induce distinct phenotypes. In terms of the total prevalence of common, rare and possible novel ACE2 missense variants, the two common variants p.Ser19Pro and p.Lys26Arg were calculated to have higher prevalence than the joint prevalence of all rare affinity variants combined and the joint prevalence of the potential novel variants are lower still. This suggests that the putative effects of most of the affinity modifying variants identified here will be isolated to individual cases and familial infection clusters. However, given the large number of infections globally, many people could still be affected and since these calculations are based only on gnomAD allele frequencies, it is possible that ACE2 affinity modifying variants may exist at higher prevalence in populations that are under-represented in gnomAD.

We compared our affinity measurements and predictions to reported SARS-CoV-2 pseudo-type infectivities against cells expressing ACE2 mutants [38]. We found a strong correlation of pseudotype infectivity with our experimental affinities and a promising association with our predicted affinities. This confirmed that affinity lowering variants protect cells from infection and establishes a "dose-response" relationship between receptor affinity and infectivity. The comparison also indicated that enhanced RBD affinity can promote infectivity, but it does not always, and the effect of further increasing affinity on infectivity rapidly diminishes. However, despite the apparently weak effect of affinity enhancing variants on infectivity, we argue that these variants could cause increased infectiousness or virulence by allowing the virus to infect a greater range of host cells and tissues. This is likely because increased entry receptor affinity is known to allow the infection of cells with decreased receptor expression in other enveloped viruses [33]. Indeed, this mechanism may be responsible for the increased infectiousness of the Spike N501Y variant, which is more effective at establishing infection in the upper airway [37] alongside greatly increased affinity for ACE2 [28].

This work is relevant to many areas of research around SARS-CoV-2 and the COVID-19 pandemic. The ACE2 variant Spike RBD affinity measurements and predictions reported here indicate candidate variants that may have SARS-CoV-2 infection and COVID-19 related phenotypes. These classifications could help efforts to develop genetic diagnostics for COVID-19

susceptibility and severity and they could guide future COVID-19 genetic association studies. Reliable affinity predictions could also be extremely useful to explore the relationship between variants in ACE2 and Spike RBD to gain new insights into SARS-CoV-2 evolution and to monitor emerging variants for potential ACE2 genotype specific risks, particularly with the common variant alleles ACE2 p.Ser19Pro and p.Lys26Arg. For emerging Spike variants, these calculations may sometimes require modelling multiple mutations on Spike RBD alongside the variants in ACE2, which will add additional uncertainty and require further strategies to validate the results. This could be achieved by comparing predictions obtained with the recently reported structures of the Alpha, Beta and Gamma variants [44] to predictions based on models of these variants derived from the ancestral Wuhan strain RBD. Finally, our general approach demonstrates the usefulness of existing computational biology technologies to address important questions in the ongoing COVID-19 pandemic, but it also highlights that it is essential to critically evaluate predictions with reference to experiments and the theoretical limitations of the underlying computational models. This is the only way to ensure that the model is relevant to the specific system being studied and to highlight any strengths and weaknesses it may have. In this case, we have shown that predictions from the state-of-the-art protein affinity prediction algorithm mCSM-PPI2[13] are useful for understanding host-virus interactions and that inhibitory variants are identified in a highly specific manner whilst there is room for improvement with respect to affinity enhancing variants.

## 4. Methods

### 4.1. ACE2 and RBD constructs

The ACE2 construct was kindly provided by Ray Owens at the Oxford Protein Production Facility-UK. The RBD construct was kindly provided by Quentin Sattentau at the Sir William Dunn School of Pathology. ACE2 point mutations were added using Agilent QuikChange II XL Site-Directed Mutagenesis Kit following the manufactures instructions. The primers were designed using the Agilent QuikChange primer design web program.

### 4.2. HEK293F suspension cell culture

Cells were grown in FreeStyle 293 Expression Medium (12338018) in a 37˚C incubator with 8% CO2 on a shaking platform at 130 rpm. Cells were passaged every 2–3 days with the suspension volume always kept below 33.3% of the total flask capacity. The cell density was kept between 0.5 and 2 million per ml.

### 4.3. Transfection of HEK293F suspension cells

Cells were counted to check cell viability was above 95% and the density adjusted to 1.0 million per ml. For 100 ml transfection, 100 μl FreeStyle MAX Reagent (16447100) was mixed with 2 ml Opti-MEM (51985034) for 5 minutes. During this incubation 100 μg of expression plasmid was mixed with 2 ml Opti-MEM. For in situ biotinylation of ACE2 90 μg of expression plasmid was mixed with 10 μg of expression plasmid encoding the BirA enzyme. The DNA was then mixed with the MAX Reagent and incubated for 25 minutes before being added to the cell culture. For ACE2 biotinylation biotin was added to the cell culture at a final concentration of 50 μM. The culture was left for 5 days for protein expression to take place.

### 4.4. Protein purification from HEK293F suspension cell supernatant

Cells were harvested by centrifugation and the supernatant collected and filtered through a 0.22 μm filter. Imidazole was added to a final concentration of 10 mM and PMSF added to a

final concentration of 1 mM. 1 ml of Ni-NTA Agarose (30310) was added per 100 ml of super-natant and the mix was left on a rolling platform at 4˚C overnight. The supernatant mix was poured through a gravity flow column to collect the Ni-NTA Agarose. The Ni-NTA Agarose was washed 3 times with 25 ml of wash buffer (50 mM NaH2PO4, 300 mM NaCl and 20 mM imidazole at pH 8). The protein was eluted from the Ni-NTA Agarose with elution buffer (50 mM NaH2PO4, 300 mM NaCl and 250 mM imidazole at pH 8). The protein was concentrated, and buffer exchanged into size exclusion buffer (25 mM NaH2PO4, 150 mM NaCl at pH 7.5) using a protein concentrator with a 10,000 molecular weight cut-off. The protein was concentrated down to less than 500 μl before loading onto a Superdex 200 10/300 GL size exclusion column. Fractions corresponding to the desired peak were pooled and frozen at -80˚C. Samples from all observed peaks were analysed on an SDS-PAGE gel (S3 Fig).

## 4.5. Surface plasmon resonance (SPR)

SARS-CoV-2 receptor binding domain binding to human extracellular ACE2 were analysed on a Biacore T200 instrument (GE Healthcare Life Sciences) at 37˚C and a flow rate of 30 μl/min. Running buffer was HBS-EP (BR100669). Streptavidin was coupled to a CM5 sensor chip (29149603) using an amine coupling kit (BR100050) to near saturation, typically 10000–12000 response units (RU). Biotinylated ACE2 WT and variants were injected into the experimental flow cells (FC2–FC4) for different lengths of time to produce desired immobilisation levels (600–700 RU). FC1 was used as a reference and contained streptavidin only. Excess streptavi-din was blocked with two 40 s injections of 250 μM biotin (Avidity). Before RBD injections, the chip surface was conditioned with 8 injections of the running buffer. A dilution series of RBD was then injected simultaneously in all FCs. Buffer was injected after every 2 or 3 RBD injections. The lowest RBD concentration was injected at the beginning and at the end of each dilution series to ensure reproducibility. The length of all injections was 30 s, and dissociation was monitored from 180–300 s. Binding measured in FC1 was subtracted from the other three FCs. Additionally, all binding and dissociation data were double referenced using the closest buffer injections [45]. In all experiments, an ACE2-specific antibody (NOVUS Biologicals, AC384) was injected at 5 μg/ml for 10 minutes with the disassociation monitored for 10 minutes (S4 Fig). Only ACE2 T27R did not bind AC384 as expected but since this mutant displays RBD binding comparable to WT ACE2, this most likely indicates direct inhibition of AC384 binding rather than the presence of unfolded protein.

## 4.6. SPR data fitting

Double referenced binding data was plotted and fit with GraphPad Prism (S5 Fig). To find the equilibrium $K_D$ (dissociation constant) the association phase was fit with a One−phase associa-tion model and the plateau binding measurements were extracted and plotted against the cor-responding concentration of RBD. This plot was then fit with a One−site specific binding model below and the value for the equilibrium $K_D$ extracted.

$$Y = \frac{Bmax * X}{(K_D + X)}$$

To convert $K_D$ values to ΔG the equation below was used. Where $K_D$ is in unit M, R is the gas constant measured in cal mol$^{-1}$ K$^{-1}$ and T is the temperature measured in K.

$$\Delta G = R * T * \ln K_D$$

A ΔΔG value could then be found for each mutant by subtracting the ΔG of the WT from the ΔG of each mutant.

For ACE2 variant D355N binding was too poor to fit accurately and, therefore, an estimate for the lower limit for the $K_D$ was calculated using the formula below. Where the "Maximum [RBD]" is the highest concentration of RBD flown over the surface, the "Binding at $K_D$ (WT)" is the RU value at the equilibrium $K_D$ for the WT protein on the same chip and the "Binding at maximum [RBD] (variant)" is the maximum RU value for the RBD binding D355N at the highest concentration. The estimated $K_D$ lower limit could then be converted into ΔG and then ΔΔG using the same method above.

$$\text{Estimated } K_D \text{ lower limit} = \text{Maximum [RBD]} * \frac{\text{Binding at } K_D \text{ (WT)}}{\text{Binding at maximum [RBD] (D355N)}}$$

## 4.7. Integration of structure, variant and mutagenesis data

The pyDRSASP suite [46] was used to integrate 3D structure, population variant and mutagenesis assay data for analysis. Population variants from gnomAD v2 [24] were mapped to ACE2 with VarAlign [46]. Residue mappings were derived from the Ensembl VEP annotations present in the gnomAD VCF. In addition, we manually checked the gnomAD multi nucleotide polymorphisms (MNPs) data file and found no records for ACE2. The structure of chimeric SARS-CoV-2 Spike receptor binding domain in complex with human ACE2 (PDB ID: 6vw1) [11] was downloaded from PDBe. Residue-residue contacts were calculated with ARPEGGIO [47]. These operations were run with our ProIntVar [46] Python package, which processes all these data into conveniently accessible Pandas DataFrames. ACE2 Spike interface residues were defined as those with any interprotein interatomic contact (S2 Table).

## 4.8. Prediction of missense variant effects on Spike–ACE2 interaction

The mCSM-PPI2[13] web server was used to predict the effect of mutations on the SARS-CoV-2 Spike-ACE2 interface topology and binding affinity with the structure PDB ID: 6vw1 [11] according to our previous protocol [23].

## 4.9. Recalibrated mCSM-PPI2 predictions with SPR tested variants

The SPR determined ΔΔG (Kd-plateau) were regressed against the mCSM-PPI2 prediction, restricting the regression to the variants with negative predicted ΔΔG, with the *lm* function in R [48]. Recalibrated scores were calculated with the *predict.lm* function and applied to ACE2 variants within 10 angstroms of Spike.

## 4.10. Enumerating possible ACE2 missense SNPs

The ACE2 gene (ENSG00000130234) was retrieved from Ensembl in Jalview [25]. Two identical CDS transcripts (ENST00000427411 and ENST00000252519) were found with the Get Cross-References command corresponding to ACE2 full-length proteins (ENSP00000389326 and ENSP00000252519). These correspond to the UniProt ACE2 sequence Q9BYF1. The CDS was saved in Fasta format and this was parsed in Python with Biopython. The CDS was broken into codons and all possible single base changes were enumerated and translated using the standard genetic code. This provided the set of amino acids accessible to each residue via a single base change.

## 4.11. Estimated Prevalence of Novel Rare Variants in ACE2 with Spike Affinity Phenotypes

**4.11.1. Upper bound from gnomAD singleton frequency (Minimum reportable frequency).** Upper bounds for the total prevalence of potential Spike affinity variants were

calculated based on the conservative assumption that novel variants must occur at lower frequencies than the minimum reportable variant frequency (i.e., minimum singleton frequency) in gnomAD. The theoretical minimum reportable frequency is the allele frequency of a singleton variant at a site where all samples have been effectively called. For alleles on the X chromosome, the proportion of XX and XY samples is important since the number of alleles sequenced is $2N_{female}+N_{male}$. Practically, many reported singleton frequencies are greater than this theoretical minimum because at a given loci not all samples have sufficient sequence data quality to be called. Therefore, the minimum reportable frequency varies by genomic position, as well as population. Since gnomAD reports the allele number (AN) only for variant sites, we used the maximum observed allele number in ACE2. For example, the maximum allele number in ACE2 corresponding to non-Finnish European samples is 80,119 (AN_NFE = 80,119). This yields a minimum observed variant frequency in ACE2 amongst non-Finnish Europeans in gnomAD (v2 exomes) of $1/80,119 = 1.2 \times 10^{-5}$, or 2.5 variants per 200,000 alleles. The total prevalence is then found by multiplying this value by the number of SNPs being considered.

**4.11.2. Empirical detection rate of rare ACE2 variants and the affinity active ratio based estimate.** Our second approach to estimate plausible frequencies of novel variants ($P$) was to project the empirical detection rate of rare ACE2 variants in gnomAD ($k$) onto the sites that correspond to the Spike interface ($n$), and then adjust this by the proportion of variants that are predicted to effect Spike binding ($\alpha$) so that,

$$P = \alpha k n$$

For example, there are 1,181 variant alleles in the 56,885 non-Finnish European cohort (AN_NFE = 80,119) arising from variants with AF < 0.01 in the 2,415 nucleotide ACE2 coding sequence (n.b. at this AF threshold, 90% are missense). This corresponds to a variation rate $k = 6.1 \times 10^{-6}$ variants per nucleotide per allele. Projecting rate $k$ onto the 25 ACE2 sites considered ($n$ = 75 nucleotides) we find $kn = 4.6 \times 10^{-4}$ variants per allele, or 91.6 variants per 200,000 alleles. Note that this should be a conservative estimate of variability at these sites, since surface residues tend to be more variable than the core or other functional regions of the protein [49], unless the site is under specific selection. The proportion of SNPs that are defined to have an affinity phenotype is dependent on the $\Delta\Delta G^{recal}$ threshold. For example, there are 38 substitutions corresponding to 41 single nucleotide variants out of the a possible 225 that are predicted to inhibit binding with $\Delta\Delta G^{recal} < -1.0$ kcal mol⁻¹ (Table 3) so that $\alpha$ = 41 / 225 = 0.18. Altogether this gives $P = 8.2 \ 10^{-5}$ for variants with $\Delta\Delta G^{recal} < -1.0$ kcal mol⁻¹. This calculation could be improved (e.g., to account for missense/ synonymous ratios in $\alpha$) but in its current form is suitable to provide estimates that indicate plausible orders of magnitude of these variants' prevalence as intended.

## 4.12. Modelling the association between experimental ΔΔG and pseudotype infectivity

Pseudotype infectivity data were taken from Shukla et al. [38] and the associated GitHub repository (http://github.com/MatreyekLab/ACE2_variants). The 'drm'function from the R *drc* package was used to fit a negative exponential model ('DRC.negExp'from package *aomisc*) to the pseudotype infectivity and SPR determined RBD affinity of ACE2 variants.

## 4.13. Software

Jalview 2.11 [25] was used for interactive sequence data retrieval, sequence analysis, structure data analysis and figure generation. UCSF Chimera [26] and PyMol [29] were used for structure analysis and figure generation.

The pyDRSASP [46] packages, comprising ProteoFAV, ProIntVar and VarAlign were used for data retrieval and analysis. Biopython was used to process sequence data.

Data analyses were coded in R and Python Jupyter Notebooks. Numpy, Pandas and Scipy were used for data analysis. Matplotlib, Seaborn and ggplot2 were used to plot data.

## Supporting information

**S1 Table. mCSM-PPI2 predictions for all gnomAD ACE2 missense variants in residues resolved in PDB 6vw1.**
(XLSX)

**S2 Table. Structural description of the ACE2 –Spike-protein interface.**
(XLSX)

**S3 Table. Association between a binary affinity classification based on predicted ΔΔG with high- and low- binding RBD binding ACE2 variants identified by deep mutagenesis.**
(PDF)

**S4 Table. Association between a 4-level affinity classification based on predicted ΔΔG with high- and low- binding RBD binding ACE2 variants identified by deep mutagenesis.**
(PDF)

**S1 Fig.** Distribution of mCSM-PPI2[13] predicted ΔΔG from in silico saturation mutagenesis of the ACE2-S interface in PDB 6vw1[11]. A. predicted ΔΔG for 475 mutations across 25 sites on ACE2 corresponding to the 23 residues within 5 Å of SARS-CoV-2 S plus Gly326 and Gly352. B. predicted ΔΔG for the subset of 151 mutations across these sites that are accessible via a single base change of the ACE2 coding sequence.
(PDF)

**S2 Fig. Comparison of SPR determined ΔΔG with the deep mutagenesis binding data from Procko and co-workers [22].** A. Experimental ΔΔG vs. ACE2 mutant enrichment in the high RBD binding population. The regression parameters are: slope = 0.38, intercept = -0.14, $R^2$ = 0.59 and p = 0.01. B. Experimental ΔΔG vs. ACE2 mutant enrichment in the low RBD binding population (negated). The regression parameters are: slope = 0.41, intercept = -0.18, $R^2$ = 0.74 and p = 0.002. Figure generated with R ggplot2.
(PDF)

**S3 Fig. Protein purification size exclusion chromatography and corresponding SDS-PAGE of labelled peaks.** A. WT RBD, B. WT ACE2, C. ACE2 mutants E37K and T27R, (continued next page) D. ACE2 mutants G326E and D355N, E. ACE2 mutants E35K, F40L and S19P, F. ACE2 mutants P329H, E329K and K26R.
(PDF)

**S4 Fig. Anti ACE2 Antibody (NOVUS AC384-NBP2-80038) binding to WT and 10 ACE2 variants.** A. Anti ACE2 binding to WT and four ACE2 variants: T27R, G326E, E37K and D355N. B. Binding to WT, S19P and K26R. C. Binding to WT, E35K and F40L. D. Binding to WT, E329G and P389H. Anti ACE2 was injected for 600 s association and buffer for a further 600 s for dissociation.
(PDF)

**S5 Fig. WT RBD binding WT ACE2 plus equilibrium KD.** A. Representative binding trace for WT RBD binding immobilised WT ACE2. Injections start from lowest to highest concentration of RBD and fits are shown in blue. B. Plateau binding values from A are plotted against the concentration of RBD injected and the equilibrium $K_D$ is extracted from the fit shown in

blue.
(PDF)

## Acknowledgments

We thank Johannes Pettmann for help with protein expression and Anna Huhn for help with data analysis. We thank Dr Jim Procter for alerting us to the release of the structure of ACE2 in complex with SARS-CoV-2 S and his initial observations regarding variants at the ACE2-S interface. We thank the Dundee Research Computing team for supporting our IT infrastructure and remote working.

## Author Contributions

**Conceptualization:** Stuart A. MacGowan, Michael I. Barton, Omer Dushek, P. Anton van der Merwe, Geoffrey J. Barton.

**Data curation:** Stuart A. MacGowan, Michael I. Barton, Mikhail Kutuzov.

**Formal analysis:** Stuart A. MacGowan, Michael I. Barton, Mikhail Kutuzov, Geoffrey J. Barton.

**Funding acquisition:** Omer Dushek, P. Anton van der Merwe, Geoffrey J. Barton.

**Investigation:** Stuart A. MacGowan, Michael I. Barton, Mikhail Kutuzov, Omer Dushek, P. Anton van der Merwe, Geoffrey J. Barton.

**Methodology:** Stuart A. MacGowan, Michael I. Barton, Mikhail Kutuzov, Omer Dushek, Geoffrey J. Barton.

**Project administration:** Omer Dushek, P. Anton van der Merwe, Geoffrey J. Barton.

**Resources:** P. Anton van der Merwe, Geoffrey J. Barton.

**Software:** Stuart A. MacGowan.

**Supervision:** Omer Dushek, P. Anton van der Merwe, Geoffrey J. Barton.

**Validation:** Stuart A. MacGowan, Michael I. Barton, Geoffrey J. Barton.

**Visualization:** Stuart A. MacGowan, Michael I. Barton.

**Writing – original draft:** Stuart A. MacGowan, Michael I. Barton.

**Writing – review & editing:** Stuart A. MacGowan, Michael I. Barton, Mikhail Kutuzov, Omer Dushek, P. Anton van der Merwe, Geoffrey J. Barton.

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
