## [Decision Letter · Decision Letter 0]

8 Oct 2021

Dear Prof Barton,

Thank you very much for submitting your manuscript "Missense variants in human ACE2 strongly affect binding to SARS-CoV-2 Spike providing a mechanism for ACE2 mediated genetic risk and protection from Covid-19" for consideration at PLOS Computational Biology.

As with all papers reviewed by the journal, your manuscript was reviewed by members of the editorial board and by several independent reviewers. In light of the reviews (below this email), we would like to invite the resubmission of a significantly-revised version that takes into account the reviewers' comments.

In particular the reference to the BioRxiv paper which I believe is now published and can therefore be updated in the manuscript.

We cannot make any decision about publication until we have seen the revised manuscript and your response to the reviewers' comments. Your revised manuscript is also likely to be sent to reviewers for further evaluation.

Sincerely,

Charlotte M Deane

Associate Editor

PLOS Computational Biology

Arne Elofsson

Deputy Editor

PLOS Computational Biology

Reviewer's Responses to Questions

**Comments to the Authors:**

Reviewer #1: In this manuscript the author investigated the contributions of missense variants on ACE2, which affect SARS-CoV-2 binding (through the RBD of the spike protein) on infectivity. First, they use deep sequencing data (created by others) and mCSM-PPI2 to calculate free energy changes of ACE2 interface mutants on binding (this part was partially published by them before in bioRxiv, in a different paper). Next, they chose 10 mutations, with either positive or negative effect on binding, or high occurrence in the population, expressed the ACE2 proteins and measured their binding with the RBD using SPR. The results of the experimental measurements were then used to recalibrate the calculations. They show that the recalibrated calculation had a better fit to the experimental data. From here, they calculated the expected change in binding affinity for all possible interface mutations. Next, they analyzed the single amino acid mutation frequency in the population, using gnomAD. The aim is to see whether carriers of different ACE2 variance will indeed have difference susceptibility to COVID19 (or to severe disease).

Comments

1. Figure 5, showing the change in binding free energy before and after recalibration of binding calculations is worrying. After calibration, it seems as if neutral mutations dominate, and the chance of getting higher and lower affinity binding mutations is almost the same. This distribution of effect of mutation on binding is different from what has been found before (see for example https://www.ncbi.nlm.nih.gov/pmc/articles/PMC4492929/). This shed doubt of the general applicability of the calibrated calculation. To test this, one should use additional mutations, not included in the experimental data from this manuscript, to test the validity of the calibrated calculations. Such mutation data are now available in a number of other recently published papers.

2. As mentioned in the paper, infectivity of pseudotype virus has been investigated for S19P and K26R, for which tighter binding was measured, but no increase in infectivity was observed. It would be helpful for this manuscript to take also the binding data published recently by Shukla (https://doi.org/10.1371/journal.ppat.1009715) and see how they fit into the calculated binding data in this manuscript. Moreover, use the infectivity data of Shukla, to compare to the binding calculation in this manuscript.

3. Shukla adds another interesting point, that the surface abundance of ACE2, which is affected by the variant, is of major importance in infectivity. Please discuss this point in the manuscript.

4. A number of variants of SARS-CoV-2 are now circulating and dominating infection (the Wuhan strain is rare now). Using binding calculations to the alpha, beta and delta variants, to ACE2 variants could provide further interesting information of the relation between potential infectivity and variants in ACE2 and SARS-CoV-2.

5. In figure 4 show both the correlation of the original and recalibrated calculation to the experimental data.

6. Why are recalibrated calculation data missing for T27R and G326E? these were the most interesting points in the original calculations.

Reviewer #2: The manuscript represents an interesting analysis of mutations in ACE2, the receptor used by the SARS-CoV-2 spike protein to enter the host cell. The specific focus is on variation with the human ACE2 gene that occurs most frequently within the large gnomAD population. The research uses the binding data from SPR experiments to recalibrate the mCSM-PPI2 specifically to predict the effect of other ACE2 variants that have not been studied by SPR. This highlights a good use of wet laboratory data to improve the accuracy of predictions to provide greater insight into variants within a specific protein.

I have the following comments:

1. The manuscript refers to previous in silico research performed. This is reference 23, which is listed as a bioRxiv preprint. It seems unusual to refer to highly related work by the same researchers that is in a preprint. Is that research under consideration for publication elsewhere? Alternatively, I wonder if it should be presented in this manuscript. I am not sure what to make of this, we are clearly in a time when there is increasing use of preprints and this is certainly to be encouraged. However, in this case it does not aid readability of the manuscript.

2. The text referring to GWAS of ACE2 variants, could be clearer, readers may not be familiar with beta etc.

3. The main text would benefit from a brief comment about the mCSM-PPI2 recalibration. I appreciate that this is covered in the methods but it would likely improve readability.

4. The paragraph that starts with “Figure 4” is very long and difficult to read. The conclusion is also one long paragraph.

5. Is the following statement in the abstract relating to use of SPR appropriate or needed? “37ÅãC, taking care to avoid common pitfalls with this method”

**Have the authors made all data and (if applicable) computational code underlying the findings in their manuscript fully available?**

Reviewer #1: Yes

Reviewer #2: Yes

PLOS authors have the option to publish the peer review history of their article (what does this mean?). If published, this will include your full peer review and any attached files.

Reviewer #1: No

Reviewer #2: No
---

## [Decision Letter · Decision Letter 1]

21 Jan 2022

Dear Prof Barton,

Thank you very much for submitting your manuscript "Missense variants in human ACE2 strongly affect binding to SARS-CoV-2 Spike providing a mechanism for ACE2 mediated genetic risk and protection from Covid-19" (PCOMPBIOL-D-21-01630R1) for consideration at PLOS Computational Biology. As with all papers peer reviewed by the journal, your manuscript was reviewed by members of the editorial board and by several independent peer reviewers. Based on the reports, we regret to inform you that we will not be pursuing this manuscript for publication at PLOS Computational Biology.

As described in the reviewers comments below there is a major concern around the the fit of the computational model to the SPR data. If all the data, including the two outliers are included the fit is bad in both original and recalibrated. In figure 4 it appears that several points are ignored, including D355A, T27R and G326E for the fitting. 

The reviews are attached below this email, and we hope you will find them helpful if you decide to revise the manuscript for submission elsewhere. We are sorry that we cannot be more positive on this occasion. We very much appreciate your wish to present your work in one of PLOS's Open Access publications. 

Thank you for your support, and we hope that you will consider PLOS Computational Biology for other submissions in the future.

Sincerely,

Charlotte M Deane

Associate Editor

PLOS Computational Biology

Arne Elofsson

Deputy Editor

PLOS Computational Biology

Reviewer's Responses to Questions

**Comments to the Authors: **

Reviewer #1: The manuscript underwent major revisions from the last version. However, it did not improve much. My major concerns are

1. As I stated before, the calculations were done on the Wuhan strain, which is not relevant. While I understand that one cannot keep up with the paste of new variants, at least, the relevance to few more recent variances should be shown. 

2. I had a better look at the Recalibrated calculations, and they did not improve what so ever the fit to the SPR data. It depends which data are used. When one takes all the data, including the two outliers the fit is bad in both original and recalibrated. In figure 4 many points are ignored, including D355A, T27R and G326E for the fitting. This does not give strong confidence to the data. 

3. The same massage, of low confidence in the calculated data can be deduced from figure 5.

4. Figure 7b, which shows all data points from the study of Shukla, one does not see a great fit of the binding data to infectivity. One sees that the lower binders may be (or not) be less infective. The tighter binders will be as infective as the Wuhan. This is expected, but gives little more specific information of specific ACE2 mutations (with very large variation between affinity and infectivity). All this was calculated for the sequence of the Wuhan strain, which is not relevant now. 

In conclusion, I dont see how this paper adds much concrete knowledge to the complicated picture of covid infectivity.

Reviewer #2: The revised manuscript addresses all of the comments from my review

**Have the authors made all data and (if applicable) computational code underlying the findings in their manuscript fully available?**

Reviewer #1: Yes

Reviewer #2: Yes

PLOS authors have the option to publish the peer review history of their article (what does this mean?). If published, this will include your full peer review and any attached files.

Reviewer #1: No

Reviewer #2: No

---

## [Editor Report · Decision Letter 2]

13 Feb 2022

Dear Prof Barton,

We are pleased to inform you that your manuscript 'Missense variants in human ACE2 strongly affect binding to SARS-CoV-2 Spike providing a mechanism for ACE2 mediated genetic risk in Covid-19: A case study in affinity predictions of interface variants' has been provisionally accepted for publication in PLOS Computational Biology.

Best regards,

Charlotte M Deane

Associate Editor

PLOS Computational Biology

Arne Elofsson

Deputy Editor

PLOS Computational Biology

---

## [Editor Report · Acceptance letter]

24 Feb 2022

PCOMPBIOL-D-21-01630R2 

Missense variants in human ACE2 strongly affect binding to SARS-CoV-2 Spike providing a mechanism for ACE2 mediated genetic risk in Covid-19: A case study in affinity predictions of interface variants

Dear Dr Barton,

I am pleased to inform you that your manuscript has been formally accepted for publication in PLOS Computational Biology. Your manuscript is now with our production department and you will be notified of the publication date in due course.

With kind regards,

Zsofia Freund
